# Medium-term outcomes of sacral neuromodulation in patients with refractory overactive bladder: A retrospective single-institution study

**Bilal Kaaki**[1,2]*, **Digant Gupta**[3]

**1** Des Moines University, Des Moines, Iowa, United States of America, **2** Department of Obstetrics and Gynecology, UnityPoint Health, Waterloo, Iowa, United States of America, **3** Clin-Science Research Consulting, Chicago, Illinois, United States of America

* Bilal.Kaaki@unitypoint.org

**Data Availability Statement:** All relevant data are within the manuscript and its Supporting Information files.

## Abstract

### Background

Sacral neuromodulation (SNM) is a minimally invasive fully reversible therapy that was approved in 1997 for overactive bladder syndrome (OAB) refractory to behavior modification and pharmacotherapy. Despite being in use for over two decades, the data on medium to long-term safety and efficacy of SNM in OAB is limited. We investigated the medium-term efficacy and safety of SNM along with the predictive factors for its success in patients with refractory OAB.

### Methods

A retrospective consecutive case series of 66 patients undergoing SNM for refractory OAB between July 2009 and July 2018. All patients underwent a test period followed by permanent implantation, if there was > = 50% improvement in any symptom. The primary outcome was "success" defined as > = 50% improvement in any clinical parameter based on the subjective assessment of patient's response. The secondary outcomes were number of pads used in 24 hours, post-operative complications and re-operation rates.

### Results

66 females with an average age of 62.7 years were included. 55/66 patients (83.3%) had a successful test phase and underwent permanent implantation. After a median follow-up of 32 months, SNM was successful in 41/55 (74.5%) patients. Mean number of pads used in 24 hours decreased significantly from 3.5 preoperatively to 1.2 at last follow-up (p<0.001). 8/55 (14.5%) patients reported complications of pain, lead migration, wound dehiscence and device malfunction. 10/55 (18.2%) patients underwent revision surgeries after a median duration of 21.9 months. Device was explanted in 15/55 (27.3%) patients after a median duration of 24 months. No significant predictor for success was identified.

**Funding:** We would like to acknowledge that the second author (Digant Gupta) of this manuscript is employed by a commercial entity called "Clin-Science Research Consulting." The funder provided support in the form of a salary for author [DG], but did not have any additional role in the study design, data collection and analysis, decision to publish, or preparation of the manuscript. The specific roles of this author are articulated in the 'author contributions' section.

**Competing interests:** The commercial affiliation "Clin-Science Research Consulting" [for DG] does not alter our adherence to PLOS ONE policies on sharing data and materials.

## Conclusions

The success rate of SNM is 75% with a complication rate of 14.5% after a median follow-up of ~3 years. This study suggests medium-term efficacy and safety but a high re-operation rate of SNM in patients with refractory OAB.

## Introduction

According to the International Continence Society (ICS), the most common types of urinary incontinence are urgency urinary incontinence (UUI), stress urinary incontinence (SUI), and mixed urinary incontinence (MUI). UUI is a common symptom of overactive bladder syndrome (OAB) which is defined by ICS as urinary urgency with or without incontinence, often accompanied by frequency (typically defined in clinical trials as $\geq 8$ micturitions per 24 hours) and nocturia (defined as $\geq 1$ micturition interrupting sleep) in the absence of urinary tract infection or other obvious pathology [1, 2]. OAB affects a significant proportion of the United States (US) population and thus represents a major public health burden [3, 4]. In one study, OAB was found to affect 42.2 million adults in the US population with an estimated disease-specific total societal cost of $24.9 billion [5]. Based on the forecasted increases in the prevalence of OAB in the US population, the total national costs in 2020 are projected to be $82.6 billion [6].

Numerous studies have assessed the prevalence of OAB in developed countries, with the National Overactive Bladder Evaluation (NOBLE) and the epidemiology of lower urinary tract symptoms (EpiLUTS) studies being the most well-known [7, 8]. NOBLE study reported the US prevalence of OAB to be 16.5%; 10.4% of OAB without UUI and 6.1% of OAB with UUI. Contrary to the popular belief, the prevalence of overall OAB was similar in women and men (16.9 vs. 16.0%) [8]. Similar estimates have been reported in the European population, with an OAB prevalence of 15.6% in men and 17.4% in women [9]. In the US-based EpiLUTS study, the prevalence of OAB depended on how OAB was defined. When symptoms were defined as "sometimes", the overall prevalence was 35.6%. With a more restrictive definition of "often", the prevalence decreased to 24.7%. Regardless of the definition, the prevalence increased with increasing age [7].

Multiple medical specialties including primary care physicians, gynecologists, urologists and subspecialists in pelvic floor disorders manage OAB in different capacities. The treatment of OAB commences with conservative therapies such as biofeedback and behavioral therapy that are often combined with second-line pharmacological treatment such as antimuscarinics and beta-3 agonists. Although behavioral and pharmacological therapies remain the first- and second-line treatments respectively for OAB, decreased efficacy and poor compliance have been reported [10–14]. A diagnosis of refractory OAB is considered when a patient has not adequately responded to lifestyle modifications, bladder training and pelvic floor therapy administered for a duration of at least 8–12 weeks, and treatment with at least one anti-muscarinic therapy for a period of at least 4–8 weeks [15]. In such patients, intra-detrusor injection of Onabotulinum toxin A, posterior tibial nerve stimulation (PTNS) and sacral neuromodulation (SNM) are the available third-line therapies [16].

SNM is a minimally invasive fully reversible therapy that was approved by the US Food and Drug Administration (FDA) in 1997 for refractory OAB. SNM is defined as a technique that electrically stimulates the third sacral spinal nerve root to modulate a neural pathway with the aim of treating bladder and/or bowel dysfunction [17]. The exact mechanism of

action of SNM is unknown, however, it is believed to correct the balance between the peripheral sacral nerves and the central nervous system by stimulating the afferent sensory fibers of the pelvic and pudendal nerves [18]. Following its approval, SNM has been increasingly utilized for OAB, and the evidence base for its medium to long-term (3 plus years) safety and efficacy is gradually building up [19–26]. However, the majority of the existing literature has been funded by the SNM device manufacturer [21, 24–29]. We therefore investigated the medium-term efficacy and safety of SNM along with the predictive factors for its success in patients with refractory OAB treated at a single institution with no funding from the manufacturer.

## Methods

### Study design and patient population

The present study was approved by the Allen College Institutional Review Board (ACIRB) and was conducted according to the guidelines laid down in the Declaration of Helsinki. Since there was no direct patient contact in this study, the need for written informed consent was waived by ACIRB. This study involved collection of existing data from patient records in such a manner that subjects cannot be identified, directly or through identifiers linked to the subjects. All patient records/information was de-identified prior to analysis.

We retrospectively examined a consecutive case series of 66 patients undergoing SNM for refractory OAB at our institution between July 2009 and July 2018. A consecutive case series of patients was chosen to avoid non-response and minimize the probability of selection bias. The inclusion criteria were: age> = 18 years, refractory OAB and a negative urinalysis. Refractory OAB was defined as symptoms of OAB and failure to respond to behavioral modifications and at least two oral pharmacologic agents. Patients with pre-existing neurological conditions affecting urinary function (eg, multiple sclerosis, myasthenia gravis, spinal cord injury, and Parkinson disease), history of pelvic radiation, and prior SNM performed for other indications such as urinary retention or fecal incontinence were excluded. Electronic medical records were reviewed to collect demographic and clinical characteristics, the presence of concomitant conditions (such as voiding symptoms, symptoms of SUI, pelvic organ prolapse [cystocele, enterocele, rectocele, uterine prolapse and vaginal vault prolapse], recurrent urinary tract infection [UTI], fecal incontinence) as well as the outcomes data.

### SNM

SNM was performed in two stages such that all patients underwent a test period (percutaneous nerve evaluation [PNE] or tined-lead evaluation [TLE]) followed by a permanent Inter-Stim II device (Medtronic Inc, Minneapolis, Minn) implantation, which was indicated only if there was a > = 50% improvement in any of the symptoms (frequency, urgency, and/or UUI). During stage 1, a test electrode (TLE) was placed percutaneously either uni- or bilaterally into the sacral foramen S3 (S2–S4) under general anesthesia or analgesic sedation without muscle relaxation, while PNE was conducted in-office under local anesthesia. A positive response to the test phase was defined as a > = 50% improvement in bladder symptoms (i.e. a drop in the number of urgency, frequency or UUI episodes by half or more), and positive responders were offered a permanent implant. During stage 2, a battery-powered implantable pulse generator (IPG) was positioned in the fatty tissue of the gluteal region on the side of the electrode and connected to the test electrode under general anesthesia or analgesic sedation [30].

## Statistical analysis

The primary outcome was "success" defined as $\geq 50\%$ improvement in any clinical parameter at the last follow-up based on the subjective assessment of patient's response by the physician as mentioned in the clinical note. In other words, success was considered if the number of urgency, frequency or UUI episodes dropped by half or more, as assessed by the physician based on patient's response. Patients also completed a global response assessment (GRA) which assessed patients' perception of overall improvement in bladder symptoms on a 5-point Likert-type scale (much worse, somewhat worse, same, somewhat better, much better). The secondary outcomes were number of pads used in 24 hours, post-operative complications and re-operation (revision and explantation) rates. The following complications were evaluated: pain, lead migration, skin erosion, seroma, hematoma, wound dehiscence, cellulitis, deep wound infection, abscess and device malfunction. The length of follow-up was calculated as the time duration between the date of permanent IPG implant and the most recent date that subjects completed follow-up.

Categorical data was summarized using frequency and percentage whereas continuous/count data using mean and standard deviation or median and range. A paired Student-t-test was used to compare the number of pads used per 24 hours at baseline and last follow-up. Logistic regression was performed to identify factors predicting the success of SNM at the last available follow-up. Owing to the retrospective nature of this study, no formal sample size calculations were conducted. All data were analyzed using IBM SPSS version 23.0 (IBM, Armonk, NY, USA). A difference was considered to be statistically significant if the p value was less than or equal to 0.05.

## Results

66 patients (all females) with a median age of 62.7 years (range 28.2–93.6 years) met the inclusion criteria. Patients' baseline characteristics are shown in Table 1. Of a total of 66 patients, 55 (83.3%) had a successful test phase and underwent permanent implantation. As a result, the final analysis included 55 patients. The median time between the test and definitive implant was 35 days (range 11–80 days). The median time to follow-up was 32 months (range 3.2–105.1 months). At the last follow-up, SNM was considered successful in 41/55 (74.5%) patients based on the subjective assessment of patient's response by the physician. The mean number of pads used in 24 hours decreased significantly from 3.5 preoperatively to 1.2 at last follow-up (p<0.001). Patients' assessment of their symptoms using GRA at the last visit were as follows: 3 (5.5%) "somewhat worse", 12 (21.8%) "same", 24 (43.6%) "somewhat better", and 16 (29.1%) "much better", indicating that 40/55 (72.7%) patients experienced improvement in bladder symptoms from their own perspective, which is consistent with an objective success rate of 74.5% reported above. None of the evaluated factors were identified as predictors of SNM success at the last follow-up visit, as shown in Table 2. Similarly, no factor was found to be associated with the occurrence of revision and explantation.

Only 8/55 (14.5%) patients experienced complications during the follow-up period such that three patients had pain, two had lead migration, one had IPG migration, one had infection/wound dehiscence, and one had device malfunction. A total of 10 (18.2%) revision surgeries were performed after a median duration of 21.9 months (range 1–77 months) from the date of permanent implant. The reasons for revisions were complications (n = 5) and end of battery life (n = 5). When end of battery life was the reason, the IPG was replaced after a median duration of 57 months (range 16–77 months) from the date of permanent implant.

Device was explanted in 15/55 (27.3%) patients after a median duration of 24 months (range 1–71 months) from the date of permanent implant. The reasons for explantation were

**Table 1. Baseline patient characteristics (N = 66).**

| Characteristic | Categories | Number (Percent) |
|---|---|---|
| Race | Caucasian | 60 (90.9) |
| | African American | 6 (9.1) |
| Menopausal status | Pre-menopausal | 8 (12.1) |
| | Post-menopausal | 58 (87.9) |
| Primary SNM indication | OAB dry | 3 (4.5) |
| | OAB wet | 63 (95.5) |
| | • *with UUI** | *28 (42.5)* |
| | • *with urgency predominant MUI** | *35 (53)* |
| Concomitant conditions | Symptoms of SUI* | 35 (53) |
| | Pelvic organ prolapse | 16 (24.2) |
| | Recurrent UTI | 11 (16.7) |
| | Voiding symptoms | 51 (77.3) |
| | Fecal incontinence | 7 (10.6) |
| | Diabetes | 19 (28.8) |
| | Chronic pelvic or back pain | 35 (53) |
| Prior treatment | Behavioral therapy | 66 (100) |
| | Anticholinergics | 66 (100) |
| | Botox | 5 (7.6) |
| SNM device | Percutaneous nerve evaluation | 62 (93.9) |
| | Tined-lead evaluation | 4 (6.1) |

| Characteristic | Mean | Median | Range |
|---|---|---|---|
| Age at baseline (years) | 62.5 | 62.7 | 28.2–93.6 |
| BMI (kg/m$^2$) at baseline | 34.6 | 34 | 18–54 |
| Parity | 2.7 | 3.0 | 0–6 |
| Symptom duration at baseline (months) | 51.1 | 36.0 | 12–180 |
| Pads used in 24 hours at baseline | 3.5 | 3.0 | 0–30 |

BMI = Body mass index; MUI = Mixed urinary incontinence; OAB = Overactive bladder syndrome; SNM = Sacral neuromodulation; SUI = Stress urinary incontinence; UTI = Urinary tract infection; UUI = Urge urinary incontinence

*UUI was defined as the complaint of involuntary loss of urine associated with urgency; SUI was defined as the complaint of involuntary loss of urine on effort or physical exertion or on sneezing or coughing; MUI was defined as the complaint of involuntary loss of urine associated with urgency and with exertion, effort, sneezing, or coughing (i.e., UUI and SUI).

decreased device efficacy (n = 5), patient undergoing magnetic resonance imaging (MRI, n = 4), pain (n = 3), end of battery life (n = 2), and infection (n = 1). None of these 15 patients got re-implanted. The 5 patients with decreased device efficacy were considered therapeutic failures and underwent explantation after a median of 24 months (range 11–29 months) from the date of permanent implant.

## Discussion

Our study, which adds to the growing evidence base on the medium-term safety and efficacy of SNM in refractory OAB, showed a success rate of 75% after a median follow-up of ~3 years. Previous studies have also evaluated the outcomes of SNM in OAB, however, these studies differ from each other with respect to the underlying sample size, data collection methodology

**Table 2. Univariate analysis of factors predicting success of SNM (N = 55).**

| Variables | OR | 95% CI | P value |
|---|---|---|---|
| Age (continuous) | 0.99 | 0.95–1.05 | 0.82 |
| BMI (continuous) | 1.06 | 0.97–1.2 | 0.21 |
| Parity (continuous) | 1.67 | 0.86–3.2 | 0.13 |
| Pads used in 24 hours at baseline (continuous) | 0.97 | 0.85–1.1 | 0.70 |
| Symptom duration in months at baseline (continuous) | 0.99 | 0.98–1.02 | 0.89 |
| Duration of test period (continuous) | 1.004 | 0.99–1.01 | 0.44 |
| Menopausal status | | | |
| Pre-menopausal (reference) | | | |
| Post-menopausal | 0.45 | 0.05–4.1 | 0.48 |
| Concomitant voiding symptoms | | | |
| No (reference) | | | |
| Yes | 0.97 | 0.22–4.2 | 0.97 |
| Concomitant symptoms of SUI | | | |
| No (reference) | | | |
| Yes | 0.64 | 0.18–2.3 | 0.49 |
| Concomitant recurrent UTI | | | |
| No (reference) | | | |
| Yes | 0.89 | 0.20–3.8 | 0.88 |
| Concomitant pelvic organ prolapse | | | |
| No (reference) | | | |
| Yes | 6.1 | 0.71–51.2 | 0.09 |
| Concomitant fecal incontinence | | | |
| No (reference) | | | |
| Yes | 0.29 | 0.05–1.6 | 0.16 |
| Concomitant diabetes | | | |
| No (reference) | | | |
| Yes | 1.9 | 0.47–8.3 | 0.35 |
| Concomitant chronic pelvic or back pain | | | |
| No (reference) | | | |
| Yes | 0.68 | 0.19–2.4 | 0.55 |

BMI = Body mass index; CI = Confidence interval; OR = Odds ratio; SNM = Sacral neuromodulation; SUI = Stress urinary incontinence; UTI = Urinary tract infection

and the length of follow-up, thereby reporting varied SNM success rates. For instance, a retrospective study by Ismail et al. conducted in 34 patients with idiopathic OAB reported a success rate of 63% after a median follow-up of 9.7 years. Additionally, 47% patients underwent revision surgeries mainly for end of battery life/device dysfunction [20]. This relatively lower success rate and higher revision rate reported by Ismail et al. could be a function of a long follow-up period. Another retrospective cohort study by Singh et al. in 65 refractory OAB patients reported a success rate of 91% and a re-operation rate of 1.5% with SNM at the end of 6 months. The higher improvement rate in the study by Singh et al. could be attributed to a shorter follow-up period as well as not using objective measurement such as voiding diary or questionnaires to define the primary outcome [31]. Siegel at al., in their prospective study of 272 patients undergoing SNM for refractory OAB, reported a 5-year success rate of 82% along with a sustained improvement in patient quality of life. The authors also reported a mean reduction of 2.0 and 5.4 leaks per day in patients with UUI and those with urgency-frequency

respectively [21]. Finally, a retrospective study by Al-zahrani et al. reported a success rate of 84.8% for urgency incontinence over a mean follow-up of 4.2 years [19]. The success rate of 75% reported in our study after a median follow-up of approximately 3 years is broadly consistent with the findings reported by other studies as described above.

We also investigated several potentially predictive factors of SNM success including voiding symptoms, symptoms of SUI, pelvic organ prolapse, diabetes, fecal incontinence and patients demographics (such as age, race and BMI), but did not find any factor to be significantly associated with SNM success. This could possibly be because of a small sample size of our study. The existing literature on the factors associated with SNM outcomes is mixed and inconclusive probably because of the differences in patient populations and types of factors evaluated across studies. Ismail et al. [20] and Starkman et al. [32] did not find any significant predictors of SNM success from amongst age, symptom duration, baseline symptom severity, presence of concomitant conditions and urodynamic parameters. On the other hand, two studies have reported age under 55 years to be significantly associated with a higher cure rate [33, 34]. While one study found SNM to be more successful in patients with more severe incontinence [22], the other reported that SNM is equally effective in treating both less severe and more severely affected groups [28]. Furthermore, one study found increasing BMI to be associated with a lower likelihood of success with SNM [35]. There is no general consensus on the factors associated with a greater probability of SNM success, and future studies with a prospective design and large sample sizes are needed to further investigate these associations.

We had a relatively low complication rate of 14.5% as compared to a rate of 30%–40% within 5 years of permanent implantation, as reported in the literature [16]. In our study, one patient had wound infection leading to dehiscence which was managed by debridement and then revision and explantation at a later date. As for any surgical procedure, strict adherence to sterile technique and emphasis on post-operative discharge instructions help decrease the infection and complication rate. Approximately, one quarter of our patients underwent explantation after a median duration of 2 years. The two main reasons for explantation were the need for MRI and decreased efficacy of the device with time. These findings should help clinicians counsel their patients about the medium-term safety and success of SNM. Moreover, our findings also demonstrate the need for an MRI-safe device in the future. In September 2019, the FDA approved a rechargeable SNM device (Axonics r-SNM System™, Irvine, CA) with a conditional safety for 1.5 Tesla full body MRI and 3 Tesla head MRI [36]. This new MRI-compatible device is expected to enable more OAB patients to choose SNM as their preferred treatment option.

We had a relatively high re-operation (revision and explantation combined) rate of 45% (18% revision rate and 27% explantation rate). We believe that the high rate of re-operation observed in our study is due to the medium-term follow up of ~3 years. The most common reasons for re-operation were end of battery life (n = 7), complications (n = 5), decreased device efficacy (n = 5) and the need for MRI (n = 4). It is important to note that the median time for explantation due to decreased device efficacy was 24 months. This finding should help clinicians counsel their patients about a potential drop in device efficacy after approximately 2 years of use.

OAB is a challenging condition to treat. Most patients stop using antimuscarinics within 6 months and only 20% continue after a year [37]. The published placebo-controlled drug trials in subjects with OAB have reported a placebo response rate ranging from 9% to 64% [38–40]. Along with Onabotulinum toxin A and PTNS, SNM has been a breakthrough in the treatment of refractory OAB and offers several advantages. A meta-analysis of 19 original studies, found Onabotulinum toxin A to be superior to placebo in reducing the episodes of urinary incontinence in patients with OAB [41]. However, intra-detrusor injection of Onabotulinum toxin A increases the risk of UTI and urinary retention necessitating clean intermittent catheterization (CIC).

PTNS is an effective neuromodulation treatment for OAB. The landmark study by Peters KM et al. demonstrated a level I evidence of its safety and efficacy in a multicenter, randomized, double-blind, sham controlled trial through 12 weeks of therapy [42]. However, with PTNS, patient compliance is a challenge since patients have to present weekly for in-office treatment, and it takes on an average more than 6 weeks for the patients to start experiencing improvement. Recently, MacDiarmid S et al. demonstrated the feasibility of an implantable nickel-sized tibial nerve stimulator in the treatment of OAB with a 71% reduction in the episodes of UUI, 70% response rate (more than 50% decrease in reported episodes of UUI) and 72% improvement in quality of life [43].

SNM overcomes the issues of UTI, urinary retention, CIC and poor compliance. However, SMN has its owns challenges that need to be addressed in the future including the need for two-staged procedures with at least one being under general anesthesia. Moreover, the implantable lead can cause complications of migration and prevent MRI use. Whether the use of SNM can lead to long-term cost reduction remains to be investigated. Future studies should also investigate therapies that are more cost-effective, can be used in-office, are less invasive without the need for general anesthesia, and are leadless.

We acknowledge limitations of our study. Despite conscious effort to monitor and control for them, this study may have been affected by biases inherent in retrospective cohort studies. The definition of primary outcome was based on the subjective assessment of patient's response by the physician without any use of objective measures such as voiding diaries, making our study susceptible to measurement bias. The sample size of 55 patients is not adequate to investigate various factors that might be associated with medium-term SNM outcomes. An absence of male subjects coupled with the fact that this study was conducted at a single institution limits the study's generalizability. As a result, additional studies from other centers and with larger sample sizes are needed to validate our findings. The strengths of this study include a homogeneous patient population with refractory OAB, relatively long median follow-up of approximately 3 years and a consecutive case series of all eligible patients during a specific time period thereby minimizing the possibility of selection bias.

## Conclusion

The success rate of SNM is 75% with a complication rate of 14.5% after a median follow-up of ~3 years. This study suggests medium-term efficacy and safety but a high re-operation rate of SNM in patients with refractory OAB. Future studies with larger sample sizes are needed to confirm these findings.

## Supporting information

**S1 Data.**
(XLSX)

## Acknowledgments

The authors would like to thank the administration of UnityPoint Health Allen Memorial Hospital for their support in providing the resources to accomplish the study.

## Author Contributions

**Conceptualization:** Bilal Kaaki.

**Data curation:** Bilal Kaaki.

**Formal analysis:** Digant Gupta.

**Investigation:** Bilal Kaaki.

**Methodology:** Bilal Kaaki, Digant Gupta.

**Supervision:** Bilal Kaaki.

**Writing – original draft:** Bilal Kaaki, Digant Gupta.

**Writing – review & editing:** Bilal Kaaki, Digant Gupta.

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
