## [Decision Letter · Decision Letter 0]

3 Jan 2020

PONE-D-19-31285

Long-term outcomes of sacral neuromodulation in patients with refractory overactive bladder: a retrospective single-institution study

PLOS ONE

Dear Dr. Gupta,

Thank you for submitting your manuscript to PLOS ONE. After careful consideration, we feel that it has merit but does not fully meet PLOS ONE’s publication criteria as it currently stands. Therefore, we invite you to submit a revised version of the manuscript that addresses the points raised during the review process.

We would appreciate receiving your revised manuscript by Feb 17 2020 11:59PM. To enhance the reproducibility of your results, we recommend that if applicable you deposit your laboratory protocols in protocols.io, where a protocol can be assigned its own identifier (DOI) such that it can be cited independently in the future. For instructions see: http://journals.plos.org/plosone/s/submission-guidelines#loc-laboratory-protocols

We look forward to receiving your revised manuscript.

Kind regards,

Andrew Zbar

Academic Editor

PLOS ONE

Additional Editor Comments (if provided):

I acted as a reviewer for this manuscript as well as the Academic Editor. I would recommend that the paper be considered after revision with several caveats suggested by the two senior reviewers with experience in SNM technology and implantation. Recommended to alter the discussion and introduction to expand on the methods of efficacy in SNM for OAB and on comparative clinical alternatives along with an expansion on the importance of the placebo effect. The authors are congratulated on an impressive set of results in the medium-term in a well written style. I would also suggest that the follow-up is medium and not long-term.

Journal Requirements:

2. We noticed you have some minor occurrence(s) of overlapping text with the following previous publication(s), which needs to be addressed:

https://doi.org/10.1007/s00192-017-3546-6

In your revision ensure you cite all your sources (including your own works), and quote or rephrase any duplicated text outside the Methods section. Further consideration is dependent on these concerns being addressed.

"The authors received no specific funding for this work."

We note that one or more of the authors are employed by a commercial company: 'Clin-Science Research Consulting'.

Reviewers' comments:

Reviewer's Responses to Questions

**Comments to the Author**

1. Is the manuscript technically sound, and do the data support the conclusions?

Reviewer #1: Partly

Reviewer #2: Yes

2. Has the statistical analysis been performed appropriately and rigorously? 

Reviewer #1: Yes

Reviewer #2: Yes

3. Have the authors made all data underlying the findings in their manuscript fully available?

Reviewer #1: No

Reviewer #2: Yes

4. Is the manuscript presented in an intelligible fashion and written in standard English?

Reviewer #1: Yes

Reviewer #2: Yes

5. Review Comments to the Author

Reviewer #1: 1. the subjective assessment of the patients by the physician : What was the relative score\\scale of the varios symptoms-frequency, urgency and UUI (how was the >50% improvment calculated)before, immediately at the end of the test period as compared to that at the follow up. Was there a change in the symptomatic expression?

2. The above information may enable to remove the column "primary SNM indication: in table 1.

3. Since there are no significant factors predicting the success rate, Table 2 can be removed leaving only the reported results in writting.

4. Complications: Pain (3) and probably decrease device efficacy(5) should be consider as complication thus raising the complication rate to 13\\55=23.6%

Reviewer #2: The authors present a retrospective analysis of the efficacy of SNM in 66 consecutively managed OAB cases with excellent results of improvement in pad use in about ¾ cases over a medium-term follow-up period. There is a moderately high SNM revision rate. Analysis did not reveal factors predictive of efficacy.

I have several caveats

1. Expand a little on the specialist groups in society affected by OAB and expand on the economic community burden in investigation and management.

2. The authors need to separate the reported incidence of occasional incontinence in women which in some studies is exceptionally high. i.e. to clarify a little their definitions in the introduction.

3. The inclusion of bladder augmentation and urinary diversion is a bit extreme and should be removed in my view.

4. Can they briefly elaborate on the proposed mechanisms of SNM in OAB?

5. Comments on funding at the end of the discussion should be placed in a funding/financial disclosure statement and not at this point.

6. Do we know anything about the clinical examination of these cases? Cystocele, rectocele, vault prolapse etc? I appreciate that these findings had no effect on efficacy.

7. The authors discuss MRI compatible implantation..they could expand on some of the rechargeable devices which have been introduced for SNM with a conditional safety for full body MRI (e.g. De Wachter et al 2019) although it is accepted that these are less a requirement for some OAB cases and more for those cases needing high battery life. At any rate there can be a little extra discussion here.

8. I would expand in the discussion on the proposed mechanisms of action of SNM in OAB (see above). I would also extend the level of sophistication of the discussion to expand on alternatives with comparison and consideration of their role e.g. PTNS, Transcutaneous TNS, peripheral implantables, transvaginal electrostimulation.

9. Some comment needs to be made regarding the placebo effect and its comparison with other forms of stimulation e.g. the SuMit Trial and others…these need to be referenced in my opinion

Overall, I enjoyed this well written article. The data are limited but impressive and the article is short and punchy with a good design. I have added some caveats to consider in a revision of this useful manuscript.

I appreciate the opportunity of reviewing this interesting article

6. PLOS authors have the option to publish the peer review history of their article (what does this mean?). If published, this will include your full peer review and any attached files.

Reviewer #1: No

Reviewer #2: Yes: Andrews P Zbar

---

## [Author Response · Author response to Decision Letter 0]

10 Feb 2020

Additional Editor Comments (if provided):

I acted as a reviewer for this manuscript as well as the Academic Editor. I would recommend that the paper be considered after revision with several caveats suggested by the two senior reviewers with experience in SNM technology and implantation. Recommended to alter the discussion and introduction to expand on the methods of efficacy in SNM for OAB and on comparative clinical alternatives along with an expansion on the importance of the placebo effect. The authors are congratulated on an impressive set of results in the medium-term in a well written style. I would also suggest that the follow-up is medium and not long-term.

MANY THANKS FOR YOUR VALUABLE FEEDBACK. WE HAVE ADDRESSED THE REVIEWERS’ COMMENTS IN THE REVISED MANUSCRIPT (PLEASE SEE OUR RESPONSES BELOW). WE HAVE ALSO CHANGED THE FOLLOW-UP DURATION FROM LONG-TERM TO MEDIUM-TERM THROUGHOUT THE MANUSCRIPT (WHERE APPLICABLE). 

Journal Requirements:

THANK YOU. WE HAVE DONE THE NEEDFUL.

2. We noticed you have some minor occurrence(s) of overlapping text with the following previous publication(s), which needs to be addressed:

https://doi.org/10.1007/s00192-017-3546-6

In your revision ensure you cite all your sources (including your own works), and quote or rephrase any duplicated text outside the Methods section. Further consideration is dependent on these concerns being addressed.

THANK YOU FOR POINTING THIS OUT. AS ADVISED, WE HAVE REPHRASED THE OVERLAPPING TEXT IN OUR REVISED MANUSCRIPT. 

"The authors received no specific funding for this work."

We note that one or more of the authors are employed by a commercial company: 'Clin-Science Research Consulting'.

AD ADVISED, WE HAVE DECLARED THE COMMERCIAL AFFILIATION (CLIN-SCIENCE RESEARCH CONSULTING) FOR THE SECOND AUTHOR. THE AUTHOR CONTRIBUTION SECTION HAS ALSO BEEN UPDATED. 

AS SUGGESTED, WE HAVE NOW INCLUDED THE AMENDED FUNDING STATEMENT IN THE COVER LETTER ACCOMPANYING THE REVISED MANUSCRIPT. 

Please also provide an updated Competing Interests Statement declaring this commercial affiliation along with any other relevant declarations relating to employment, consultancy, patents, products in development, or marketed products, etc. 

AS ADVISED, WE HAVE CONFIRMED IN THE COVER LETTER THAT THE COMMERCIAL AFFILIATION DOES NOT ALTER OUR ADHERENCE TO PLOS ONE POLICIES ON SHARING DATA AND MATERIALS.

AS ADVISED, WE HAVE INCLUDED BOTH AN UPDATED FUNDING STATEMENT AND COMPETING INTERESTS STATEMENT IN OUR COVER LETTER. WE WOULD BE GRATEFUL IF YOU COULD PLEASE UPDATE THE ONLINE SUBMISSION FORM ON OUR BEHALF. THANK YOU. 

Review Comments to the Author

Reviewer #1: 1. The subjective assessment of the patients by the physician : What was the relative score\\scale of the various symptoms-frequency, urgency and UUI (how was the >50% improvement calculated) before, immediately at the end of the test period as compared to that at the follow up. Was there a change in the symptomatic expression?

THANK FOR THIS QUERY. SUCCESS WAS DEFINED AS >=50% IMPROVEMENT IN ANY CLINICAL PARAMETER AT THE LAST FOLLOW-UP BASED ON THE SUBJECTIVE ASSESSMENT OF PATIENT’S RESPONSE BY THE PHYSICIAN AS MENTIONED IN THE CLINICAL NOTE. IN OTHER WORDS, SUCCESS WAS CONSIDERED IF THE NUMBER OF URGENCY, FREQUENCY OR UUI EPISODES DROPPED BY HALF OR MORE, AS ASSESSED BY THE PHYSICIAN BASED ON PATIENT’S RESPONSE. 

THE ABOVE HAS BEEN CLARIFIED IN THE METHODS SECTION OF THE REVISED MANUSCRIPT. 

2. The above information may enable to remove the column "primary SNM indication: in table 1.

THANK YOU FOR THIS SUGGESTION. HOWEVER, WE WOULD LIKE TO REQUEST RETAINING THIS COLUMN IN THE TABLE BECAUSE IT DESCRIBES THAT ONLY 3 PATIENTS HAD URGENCY-FREQUENCY SYMPTOMS WITH NO INCONTINENCE (I.E. DRY OAB), WHEREAS 63 HAD URGENCY INCONTINENCE (I.E. WET OAB). OUT OF THE 63 WET OAB PATIENTS, 35 HAD MIXED URINARY INCONTINENCE WITH URGENCY PREDOMINANCE AND 28 HAD URGENCY URINARY INCONTINENCE.

3. Since there are no significant factors predicting the success rate, Table 2 can be removed leaving only the reported results in writing.

WE UNDERSTAND THAT NONE OF THE EVALUATED FACTORS WERE SIGNIFICANT IN PREDICTING THE SUCCESS, HOWEVER WE WOULD STILL REQUEST TO RETAIN TABLE 2 IN THE MANUSCRIPT, BECAUSE THE TABLE PROVIDES THE PARAMETER ESTIMATES (OR ODDS RATIOS) FOR DIFFERENT RISK FACTORS WHICH MIGHT BE HELPFUL FOR OTHER RESEARCHERS IN THEIR FUTURE INVESTIGATIONS. 

4. Complications: Pain (3) and probably decrease device efficacy (5) should be consider as complication thus raising the complication rate to 13\\55=23.6%

AS SUGGESTED, WE HAVE INCLUDED PAIN AS A COMPLICATION, SO THE NEW COMPLICATION RATE IS NOW 8/55=14.5%. THIS HAS BEEN UPDATED THROUGHOUT THE MANUSCRIPT. 

DECREASED DEVICE EFFICACY (N=5) WAS NOT A DIRECT COMPLICATION OF THE IMPLANT PROCEDURE AND CAN PARTIALLY BE CONTRIBUTED BY A PHYSIOLOGIC EFFECT DUE TO HABITUATION OF THE NERVOUS SYSTEM. AS A RESULT, WE HAVE NOT CONSIDERED DECREASED DEVICE EFFICACY AS A COMPLICATION. 

Reviewer #2: The authors present a retrospective analysis of the efficacy of SNM in 66 consecutively managed OAB cases with excellent results of improvement in pad use in about ¾ cases over a medium-term follow-up period. There is a moderately high SNM revision rate. Analysis did not reveal factors predictive of efficacy.

I have several caveats:

1. Expand a little on the specialist groups in society affected by OAB and expand on the economic community burden in investigation and management.

AS SUGGESTED, THIS DESCRIPTION HAS NOW BEEN ADDED IN THE INTRODUCTION SECTION OF THE REVISED MANUSCRIPT. APPROPRIATE REFERENCES HAVE ALSO BEEN ADDED. 

2. The authors need to separate the reported incidence of occasional incontinence in women which in some studies is exceptionally high. i.e. to clarify a little their definitions in the introduction.

AS ADVISED, WE HAVE ADDED MORE CLARITY ON THE DIFFERENT TYPES OF URINARY INCONTINENCE IN THE INTRODUCTION SECTION. PLEASE NOTE THAT OUR STUDY WAS FOCUSED ON PATIENTS WITH REFRACTORY OAB. 

3. The inclusion of bladder augmentation and urinary diversion is a bit extreme and should be removed in my view.

AS ADVISED, WE HAVE REMOVED THESE TREATMENT OPTIONS FROM THE INTRODUCTION. 

4. Can they briefly elaborate on the proposed mechanisms of SNM in OAB?

AS ADVISED, THIS HAS NOW BEEN DONE IN THE INTRODUCTION SECTION OF THE REVISED MANUSCRIPT. APPROPRIATE REFERENCES HAVE ALSO BEEN ADDED. 

5. Comments on funding at the end of the discussion should be placed in a funding/financial disclosure statement and not at this point.

AS SUGGESTED, WE HAVE REMOVED THE FUNDING STATEMENT FROM THE DISCUSSION SECTION. AS A PART OF THE JOURNAL REQUIREMENT, WE HAVE NOW INCLUDED THE AMENDED FUNDING STATEMENT IN THE COVER LETTER ACCOMPANYING THE REVISED MANUSCRIPT. 

6. Do we know anything about the clinical examination of these cases? Cystocele, rectocele, vault prolapse etc? I appreciate that these findings had no effect on efficacy.

THE FOLLOWING CONCOMITANT CONDITIONS WERE EVALUATED: VOIDING DYSFUNCTION, STRESS URINARY INCONTINENCE [SUI], PELVIC ORGAN PROLAPSE, RECURRENT URINARY TRACT INFECTION [UTI], AND FECAL INCONTINENCE. WITHIN THE CATEGORY OF PELVIC ORGAN PROLAPSE, THE FOLLOWING CONDITIONS WERE INCLUDED: CYSTOCELE, ENTEROCELE, RECTOCELE, UTERINE PROLAPSE AND VAGINAL VAULT PROLAPSE. THIS HAS NOW BEEN CLARIFIED IN THE METHODS SECTION OF THE REVISED MANUSCRIPT. 

7. The authors discuss MRI compatible implantation…they could expand on some of the rechargeable devices which have been introduced for SNM with a conditional safety for full body MRI (e.g. De Wachter et al 2019) although it is accepted that these are less a requirement for some OAB cases and more for those cases needing high battery life. At any rate there can be a little extra discussion here.

AS ADVISED, WE HAVE NOW INCLUDED SOME DISCUSSION AROUND THIS NEW DEVICE IN THE REVISED MANUSCRIPT. APPROPRIATE REFERENCE HAS ALSO BEEN ADDED. 

8. I would expand in the discussion on the proposed mechanisms of action of SNM in OAB (see above). I would also extend the level of sophistication of the discussion to expand on alternatives with comparison and consideration of their role e.g. PTNS, Transcutaneous TNS, peripheral implantables, transvaginal electrostimulation.

AS ADVISED, WE HAVE NOW EXPANDED THE DISCUSSION ON THE ROLE OF SOME TREATMENT ALTERNATIVES FOR OAB (ALONG WITH THE SUPPORTING REFERENCES). 

9. Some comment needs to be made regarding the placebo effect and its comparison with other forms of stimulation e.g. the SuMit Trial and others…these need to be referenced in my opinion

AS ADVISED, A DISCUSSION ON PACEBO EFFECT (ALONG WITH SUMIT TRIAL AND OTHER RELEVANT STUDIES) HAS BEEN INCLUDED IN THE REVISED MANUSCRIPT. 

Overall, I enjoyed this well written article. The data are limited but impressive and the article is short and punchy with a good design. I have added some caveats to consider in a revision of this useful manuscript.

I appreciate the opportunity of reviewing this interesting article.

THANK YOU FOR ALL YOUR CONSTRUCTIVE FEEDBACK IN HELPING US IMPROVE OUR MANUSCRIPT.

---

## [Decision Letter · Decision Letter 1]

4 Jun 2020

PONE-D-19-31285R1

Medium-term outcomes of sacral neuromodulation in patients with refractory overactive bladder: a retrospective single-institution study

PLOS ONE

Dear Dr. Gupta,

Thank you for submitting your manuscript to PLOS ONE. After careful consideration, we feel that it has merit but does not fully meet PLOS ONE’s publication criteria as it currently stands. Therefore, we invite you to submit a revised version of the manuscript that addresses the points raised during the review process.

ACADEMIC EDITOR: I take this opportunity to point out to you that the origin of 'overactive bladder' in the ICS is not overactive bladder, but 'overactive bladder syndrome' and ask you to correct this. You have also introduced 'voiding dysfunction' and 'SUI' in the manuscript. Your manuscript does however not state that it has been established other than on the basis of symptoms. That is why I ask you to think about replacing with 'voiding symptoms' and 'symptoms of stress urinary incontinence'. You also seem, as a reaction to earlier reviews, to be concerned about the numbers of patients with the syndrome and the costs involved, I agree with that. Can you include in the discussion how your approach - with an expensive intervention, without any kind of objective diagnosis - can lead to a reduction in costs and whether, for example, how medical self-regulation can be used to reduce costs?

We look forward to receiving your revised manuscript.

Kind regards,

Peter F.W.M. Rosier, M.D. PhD

Academic Editor

PLOS ONE

Reviewers' comments:

Reviewer's Responses to Questions

**Comments to the Author**

1. If the authors have adequately addressed your comments raised in a previous round of review and you feel that this manuscript is now acceptable for publication, you may indicate that here to bypass the “Comments to the Author” section, enter your conflict of interest statement in the “Confidential to Editor” section, and submit your "Accept" recommendation.

Reviewer #1: All comments have been addressed

Reviewer #3: (No Response)

2. Is the manuscript technically sound, and do the data support the conclusions?

Reviewer #1: Yes

Reviewer #3: Partly

3. Has the statistical analysis been performed appropriately and rigorously? 

Reviewer #1: Yes

Reviewer #3: Yes

4. Have the authors made all data underlying the findings in their manuscript fully available?

Reviewer #1: Yes

Reviewer #3: No

5. Is the manuscript presented in an intelligible fashion and written in standard English?

Reviewer #1: Yes

Reviewer #3: Yes

6. Review Comments to the Author

Reviewer #1: The authors have answered the reviewers comments. It is a detailed though retrospective work which reasures physicians as well as patients with ORB to use SNM as a sound therapeutic option. The work is now eligible for publication

Reviewer #3: The authors have a nice retrospective case series with 66 patients having a median follow up of 32 months. This is not long-term but they have had excellent outcomes with 74.5% reporting >50% improvement.

They report 27.3% explant at 24 months. This is high, even for 32 months’ follow up, when compared to 19.1% by Siegel et al who report 5 year outcomes for a multicenter RCT (INSITE)[1]. IT should be clarified by the authors how many of these 27% got re-implanted when appropriate and for which reasons. The 5 explanted for loss of efficacy should be specifically stated for the reader if they were counted as therapeutic failures.

The authors make the statement that medium and long term outcomes for safety and efficacy for SNM are limited. This is not really supported by the literature [1-9] with outcomes as long as 17 years for tined lead removal [6] and one manuscript dating back to 2010 with 5 year average outcomes [5].

PLOS One requires that all data be made available to the reader, it is not part of the submitted manuscript and as far as I can tell, not submitted as supporting information. This should be remedied as a requirement of the journal.

Table 1, it is a little confusing that the authors report mean followed by a comma, then median results. This would normally be reported as 2 separate columns and I suspect that will be an easy change and make it easier for the reader to understand

Table 2, for what it’s worth I agree with the authors’ response to reviewer #1 that although no factors were significant predictors, calling attention to this with a table may be helpful to some readers. I think publishing negative results can be nearly as helpful as positive ones in some cases.

Overall this appears to be a community based medium-term case series and may be of interest to some readers for that reason.

[1] Siegel S, Noblett K, Mangel J, Bennett J, Griebling TL, Sutherland SE, et al. Five-Year Followup Results of a Prospective, Multicenter Study of Patients with Overactive Bladder Treated with Sacral Neuromodulation. J Urol. 2018;199:229-36.

[2] Amundsen CL, Komesu YM, Chermansky C, Gregory WT, Myers DL, Honeycutt EF, et al. Two-Year Outcomes of Sacral Neuromodulation Versus OnabotulinumtoxinA for Refractory Urgency Urinary Incontinence: A Randomized Trial. Eur Urol. 2018;74:66-73.

[3] Blok B, Van Kerrebroeck P, de Wachter S, Ruffion A, Van der Aa F, Perrouin-Verbe MA, et al. Two-year safety and efficacy outcomes for the treatment of overactive bladder using a long-lived rechargeable sacral neuromodulation system. Neurourol Urodyn. 2020.

[4] Ismail S, Chartier-Kastler E, Perrouin-Verbe MA, Rose-Dite-Modestine J, Denys P, Phe V. Long-Term Functional Outcomes of S3 Sacral Neuromodulation for the Treatment of Idiopathic Overactive Bladder. Neuromodulation. 2017;20:825-9.

[5] Marinkovic SP, Gillen LM, Marinkovic CM. Minimum 6-year outcomes for interstitial cystitis treated with sacral neuromodulation. Int Urogynecol J. 2011;22:407-12.

[6] Powell CR, Kreder KJ. Long-term outcomes of urgency-frequency syndrome due to painful bladder syndrome treated with sacral neuromodulation and analysis of failures. J Urol. 2010;183:173-6.

[7] Rueb JJ, Pizarro-Berdichevsky J, Goldman HB. 17-Year Single Center Retrospective Review of Rate, Risk Factors, and Outcomes of Lead Breakage during Sacral Neuromodulation Lead Removal. J Urol. 2020:101097JU0000000000000740.

[8] Yazdany T, Bhatia N, Nguyen J. Determining outcomes, adverse events, and predictors of success after sacral neuromodulation for lower urinary disorders in women. Int Urogynecol J. 2011;22:1549-54.

[9] Zegrea A, Kirss J, Pinta T, Rautio T, Varpe P, Kairaluoma M, et al. Outcomes of sacral neuromodulation for chronic pelvic pain: a Finnish national multicenter study. Tech Coloproctol. 2020;24:215-20.

7. PLOS authors have the option to publish the peer review history of their article (what does this mean?). If published, this will include your full peer review and any attached files.

Reviewer #1: No

Reviewer #3: Yes: CR Powell, MD

---

## [Author Response · Author response to Decision Letter 1]

7 Jun 2020

ATTACHED BELOW, FOR YOUR PERUSAL, IS A DETAILED DESCRIPTION (HIGHLIGHTED IN RED AND CAPS) OF HOW WE’VE ADDRESSED THE REVIEWERS’ COMMENTS.

Additional Editor Comments:

I take this opportunity to point out to you that the origin of 'overactive bladder' in the ICS is not overactive bladder, but 'overactive bladder syndrome' and ask you to correct this. 

THANKS FOR POINTING THIS OUT. AS SUGGESTED, WE HAVE MADE THIS CORRECTION IN THE REVISED MANUSCRIPT (BOTH IN THE ABSTRACT AND THE MAIN TEXT). 

You have also introduced 'voiding dysfunction' and 'SUI' in the manuscript. Your manuscript does however not state that it has been established other than on the basis of symptoms. That is why I ask you to think about replacing with 'voiding symptoms' and 'symptoms of stress urinary incontinence'. 

WE HAVE MADE THE SUGGESTED REPLACEMENTS THROUGHOUT THE MANUSCRIPT. 

You also seem, as a reaction to earlier reviews, to be concerned about the numbers of patients with the syndrome and the costs involved, I agree with that. Can you include in the discussion how your approach - with an expensive intervention, without any kind of objective diagnosis - can lead to a reduction in costs and whether, for example, how medical self-regulation can be used to reduce costs?

THE FOCUS OF OUR PAPER WAS NOT ON THE COST-BENEFIT ANALYSIS OF SNM, HOWEVER, BASED ON YOUR SUGGESTION, WE HAVE INCLUDED A GENERAL POINT IN THE DISCUSSION THAT THE LONG-TERM COST BENEFITS OF SNM NEED TO BE INVESTIGATED IN FUTURE STUDIES. 

Review Comments to the Author

Reviewer #1: The authors have answered the reviewers’ comments. It is a detailed though retrospective work which reassures physicians as well as patients with ORB to use SNM as a sound therapeutic option. The work is now eligible for publication.

MANY THANKS ONCE AGAIN FOR YOUR VALUABLE FEEDBACK. 

Reviewer #3: The authors have a nice retrospective case series with 66 patients having a median follow up of 32 months. This is not long-term but they have had excellent outcomes with 74.5% reporting >50% improvement.

THANK YOU.

They report 27.3% explant at 24 months. This is high, even for 32 months’ follow up, when compared to 19.1% by Siegel et al who report 5 year outcomes for a multicenter RCT (INSITE)[1]. It should be clarified by the authors how many of these 27% got re-implanted when appropriate and for which reasons. NONE OF THESE 27% (N=15) PATIENTS GOT RE-IMPLANTED. THIS HAS BEEN CLARIFIED IN THE RESULTS SECTION OF THE REVISED MANUSCRIPT. The 5 explanted for loss of efficacy should be specifically stated for the reader if they were counted as therapeutic failures. YES, THEY WERE CONSIDERED THERAPEUTIC FAILURES. THIS HAS BEEN CLARIFIED IN THE RESULTS SECTION OF THE REVISED MANUSCRIPT. 

The authors make the statement that medium and long term outcomes for safety and efficacy for SNM are limited. This is not really supported by the literature [1-9] with outcomes as long as 17 years for tined lead removal [6] and one manuscript dating back to 2010 with 5 year average outcomes [5].

THANK YOU FOR BRINGING THESE ADDITIONAL REFERENCES TO OUR ATTENTION. WE HAVE NOW INCLUDED A FEW OF THOSE IN THE INTRODUCTION SECTION AND HAVE ALSO MODIFIED THE STATEMENT REGARDING LIMITED DATA. 

PLOS One requires that all data be made available to the reader, it is not part of the submitted manuscript and as far as I can tell, not submitted as supporting information. This should be remedied as a requirement of the journal.

THANKS FOR RAISING THIS POINT. PLEASE NOTE THAT AS PER THE JOURNAL REQUIREMENTS, WE HAVE INCLUDED THE FOLLOWING IN OUR COVER LETTER TO THE EDITOR. 

We would like to confirm that the data underlying the findings in our study cannot be made publicly available due to ethical and legal restrictions. The original data contains potentially identifying or sensitive patient information. This restriction has been imposed by the Allen College Institutional Review Board (ACIRB). However, a copy of the de-identified dataset underlying the analyses reported in this paper is available to all interested researchers upon request to the ACIRB (contact person is Lisa Brodersen who can be reached at Lisa.Brodersen@allencollege.edu or 319-226-2000).

Table 1, it is a little confusing that the authors report mean followed by a comma, then median results. This would normally be reported as 2 separate columns and I suspect that will be an easy change and make it easier for the reader to understand

AS ADVISED, WE HAVE REPORTED MEANS AND MEDIANS IN 2 SEPARATE COLUMNS. 

Table 2, for what it’s worth I agree with the authors’ response to reviewer #1 that although no factors were significant predictors, calling attention to this with a table may be helpful to some readers. I think publishing negative results can be nearly as helpful as positive ones in some cases.

WE AGREE WITH YOUR THOUGHTS ON THIS. AS A RESULT, WE HAVE RETAINED TABLE 2 IN OUR REVISED MANUSCRIPT. 

Overall this appears to be a community based medium-term case series and may be of interest to some readers for that reason.

THANKS AGAIN FOR TAKING THE TIME TO REVIEW OUR MANUSCIPT AND PROVIDING YOUR VALUABLE COMMENTS. 

[1] Siegel S, Noblett K, Mangel J, Bennett J, Griebling TL, Sutherland SE, et al. Five-Year Followup Results of a Prospective, Multicenter Study of Patients with Overactive Bladder Treated with Sacral Neuromodulation. J Urol. 2018;199:229-36.

[2] Amundsen CL, Komesu YM, Chermansky C, Gregory WT, Myers DL, Honeycutt EF, et al. Two-Year Outcomes of Sacral Neuromodulation Versus OnabotulinumtoxinA for Refractory Urgency Urinary Incontinence: A Randomized Trial. Eur Urol. 2018;74:66-73.

[3] Blok B, Van Kerrebroeck P, de Wachter S, Ruffion A, Van der Aa F, Perrouin-Verbe MA, et al. Two-year safety and efficacy outcomes for the treatment of overactive bladder using a long-lived rechargeable sacral neuromodulation system. Neurourol Urodyn. 2020.

[4] Ismail S, Chartier-Kastler E, Perrouin-Verbe MA, Rose-Dite-Modestine J, Denys P, Phe V. Long-Term Functional Outcomes of S3 Sacral Neuromodulation for the Treatment of Idiopathic Overactive Bladder. Neuromodulation. 2017;20:825-9.

[5] Marinkovic SP, Gillen LM, Marinkovic CM. Minimum 6-year outcomes for interstitial cystitis treated with sacral neuromodulation. Int Urogynecol J. 2011;22:407-12.

[6] Powell CR, Kreder KJ. Long-term outcomes of urgency-frequency syndrome due to painful bladder syndrome treated with sacral neuromodulation and analysis of failures. J Urol. 2010;183:173-6.

[7] Rueb JJ, Pizarro-Berdichevsky J, Goldman HB. 17-Year Single Center Retrospective Review of Rate, Risk Factors, and Outcomes of Lead Breakage during Sacral Neuromodulation Lead Removal. J Urol. 2020:101097JU0000000000000740.

[8] Yazdany T, Bhatia N, Nguyen J. Determining outcomes, adverse events, and predictors of success after sacral neuromodulation for lower urinary disorders in women. Int Urogynecol J. 2011;22:1549-54.

[9] Zegrea A, Kirss J, Pinta T, Rautio T, Varpe P, Kairaluoma M, et al. Outcomes of sacral neuromodulation for chronic pelvic pain: a Finnish national multicenter study. Tech Coloproctol. 2020;24:215-20.

---

## [Decision Letter · Decision Letter 2]

22 Jun 2020

PONE-D-19-31285R2

Medium-term outcomes of sacral neuromodulation in patients with refractory overactive bladder: a retrospective single-institution study

PLOS ONE

Dear Dr. Gupta,

Thank you for submitting your manuscript to PLOS ONE. After careful consideration, we feel that it has merit but does not fully meet PLOS ONE’s publication criteria as it currently stands. Therefore, we invite you to submit a revised version of the manuscript that addresses the points raised during the review process.

ACADEMIC EDITOR:

This area is saturated in the literature and the work provides not much news. It is not easy to decide positive for publication if you do not make the data publicly available, which is a PLOS ONE requirement. When there are ethical considerations, they are not different when data is available only 'on request'.

We look forward to receiving your revised manuscript.

Kind regards,

Peter F.W.M. Rosier, M.D. PhD

Academic Editor

PLOS ONE

Reviewers' comments:

Reviewer's Responses to Questions

**Comments to the Author**

1. If the authors have adequately addressed your comments raised in a previous round of review and you feel that this manuscript is now acceptable for publication, you may indicate that here to bypass the “Comments to the Author” section, enter your conflict of interest statement in the “Confidential to Editor” section, and submit your "Accept" recommendation.

Reviewer #3: All comments have been addressed

2. Is the manuscript technically sound, and do the data support the conclusions?

Reviewer #3: Yes

3. Has the statistical analysis been performed appropriately and rigorously? 

Reviewer #3: No

4. Have the authors made all data underlying the findings in their manuscript fully available?

Reviewer #3: Yes

5. Is the manuscript presented in an intelligible fashion and written in standard English?

Reviewer #3: Yes

6. Review Comments to the Author

Reviewer #3: The work is not novel but is reasonable. it does not increase the literature which is saturated in this area.

7. PLOS authors have the option to publish the peer review history of their article (what does this mean?). If published, this will include your full peer review and any attached files.

Reviewer #3: No

---

## [Author Response · Author response to Decision Letter 2]

22 Jun 2020

This area is saturated in the literature and the work provides not much news. It is not easy to decide positive for publication if you do not make the data publicly available, which is a PLOS ONE requirement. When there are ethical considerations, they are not different when data is available only 'on request'.

AS SUGGESTED, WE HAVE MADE THE DATA PUBLICLY AVAILABLE. A COPY OF THE DE-IDENTIFIED DATASET IN MS EXCEL FORMAT HAS BEEN UPLOADED TO THE SUBMISSION SYSTEM.

---

## [Editor Report · Decision Letter 3]

26 Jun 2020

Medium-term outcomes of sacral neuromodulation in patients with refractory overactive bladder: a retrospective single-institution study

PONE-D-19-31285R3

Dear Dr. Gupta,

We’re pleased to inform you that your manuscript has been judged scientifically suitable for publication and will be formally accepted for publication once it meets all outstanding technical requirements.

Kind regards,

Peter F.W.M. Rosier, M.D. PhD

Academic Editor

PLOS ONE
---

## [Editor Report · Acceptance letter]

30 Jun 2020

PONE-D-19-31285R3 

Medium-term outcomes of sacral neuromodulation in patients with refractory overactive bladder: a retrospective single-institution study 

Dear Dr. Gupta:

I'm pleased to inform you that your manuscript has been deemed suitable for publication in PLOS ONE. Congratulations! Your manuscript is now with our production department. 

Kind regards, 

on behalf of

Dr. Peter F.W.M. Rosier 

Academic Editor

PLOS ONE